# Algorithm and Examples of an Agent-Based Evacuation Model

**Xiaoting Cui [1], Jingwei Ji [2,*] and Xuehe Bai [2]**

[1]  Nanjing University of Science and Technology, Nanjing 210094, China
[2]  Jiangsu Key Laboratory of Fire Safety in Urban Underground Space, China University of Mining and Technology, Xuzhou 221000, China
*  Correspondence: jjwcn@cumt.edu.cn

**Abstract:** This research establishes a "detect-decide-action" agent-based evacuation model based on the social force model, introducing an active steering force into the basis of the dynamic equation with the combination of the behavioral decision model and the probability model. In the AEM, the detection algorithm is used to identify pedestrians or obstacles within the detection radius to provide the next walking direction and apply the active steering force. The obstacle avoidance algorithm is the core of the "action" link. This research focuses on the establishment of the following and bypassing algorithm when moving in the same direction, and the algorithm of a detour when moving in the opposite direction, applying C++ programming language to achieve the basic evacuation behavior simulation of avoiding pedestrians and obstacles in the actual scene. The results show that compared with the grid model and the general social force model, the agent model (AEM) solves the problem of the distortion of evacuation behavior to some extent, and the pedestrian is more flexible in the choice of evacuation path.

**Keywords:** agent-based evacuation model; decision making; obstacle avoidance algorithm





## 1. Introduction

With the rapid development of the economy, the number of large-scale building sites is increasing, and the building structure is also developing to the upper space (high-rise) and the lower space (underground). These building sites are often densely populated and have complex human flow organization, so how to effectively manage the crowd and carry out efficient evacuation in the event of an emergency has attracted more and more attention from researchers [1]. The research on pedestrian groups is mainly carried out through direct observation and evacuation experiments, but the time and resources spent on evacuation experiments and observation are relatively large, so it is difficult to extract relevant observation data, and there are certain limitations in repeatability and operability [2,3]. So in some cases, engineers use back-of-the-envelope calculations to assess life safety, and, in other cases, computerized evacuation models [4]. These are various practical and effective methods used to establish a pedestrian evacuation dynamic model and carry out computer simulation [5,6]. The psychological tendency of pedestrians sliding downhill and practicing collision avoidance behavior is considered for simulation modeling. Numerous evacuation models established by previous scholars can be generally categorized into macro and micro models by the characteristics of involving evacuees [7,8].

The macro model regards evacuees as individuals with the same characteristics, considering the evacuation as a movement of uniform flow, and the density distribution of evacuated people is an important factor affecting evacuation. The macro model studies the evacuation system from the overall perspective and focuses on the relationship between the macro statistical parameters (speed, flow, density) of individuals' flow. Fruin [9] (1971) made a comprehensive statistical analysis of crowd movement in different scenes and put forward a model to describe the evacuation rule of evacuees by using aerodynamics. The model analogized the crowd as a fluid, applying the hydrodynamic equation to determine

the change in crowd density and speed with time [10,11]. Henderson [12] also believed that group motion was similar to gas and liquid motion with the property of moving fluid mechanics. On this basis, he proposed a fluid dynamics model. The principle of the model was similar to the Boltzmann equation, simulating the process of group evacuation. Lovas [13], Thompson [14], Watts [15], and other researchers brought up a queuing network model [16] as a tool to simulate the evacuation of building fire, mainly supporting the study of evacuation capacity in traffic places such as subway stations [17]. The macro evacuation model is suitable for large-scale crowd simulation, but it neglects the interaction between evacuees and hardly describes the interaction phenomenon during the evacuation process, leading to the distortion of the evacuation process and results to some extent.

On the contrary, the micro model distinguishes the involving evacuees who have their own attributes, such as speed, direction, response time, etc., which shows the overall movement of the population through the interaction between individuals, as well as between individuals and obstacles, including the cellular automata model, the lattice gas model, the social force model, and the agent-based model.

The cellular automata model [6,18] and the lattice gas model [19–21] can discretize the attributes of time, space, and pedestrian movement; synchronize the movement of many pedestrians in limited time and space; and simulate the whole evacuation process. However, the direction of pedestrian flow is constrained by spatial discretization. The main limitation of these micro models lies in their over-discretization in space and time.

Lewin [22] came up with the theory that the change in pedestrian behavior was caused by social forces, which means that it was feasible that the pedestrian behavior rules could be transformed into motion equations; based on this, Helbing [23,24] put forward the initial idea of micro modeling after studying the multitudes of pedestrian behavior models and enlightening different vehicle traffic micro methods via the social force model [25]. The model divided the forces of a pedestrian during the process of evacuation into three parts: the original driving force, the forces between pedestrians, and the forces between pedestrians and obstacles. The social force model was used to simulate many observed phenomena. Evacuees rely on the force they receive to maintain their progress and their moving direction. In various practical scenes, during the process of evacuation, there might be a situation where the path driven by forces is not consistent with the optimal path, and evacuees will be led to a longer path, or where the forces are balanced so that the pedestrian will stay still. This mechanical model makes it difficult to explain the behaviors of following, waiting, or surpassing.

The agent-based model is able to adjust its state continuously to make decisions according to the goal set by itself or the system. It was first used in artificial intelligence and its related fields [26]. With the gradual maturity of the agent theory and the cultivation of science and technology, the application has been expanded to the simulation of crowd evacuation in various circumstances. The virtual agent is implied to simulate the movement of individuals in the actual scene to obtain relevant feedback information to formulate the corresponding measures according to the specified code of conduct [27]. The agent-based model has a high fidelity of similarity with the actual characteristics of the pedestrian's motion [28]. T Korhonen proposes an evacuation simulation method in which each agent has its own individual attributes and embeds the method in a CFD-based fire modeling program [29], and C. Natalie van der Wal created an agent-based model of an evacuating crowd. Socio-cultural factors that were modeled are familiarity with the environment, response time, and the social contagion of fear and beliefs about the situation [30].

To realize the intellectualization of pedestrian evacuation behavior, this research establishes an agent-based evacuation model combining the mechanical model and the decision-making model. Different from the single-based mechanical social force model, the AEM constructs a real virtual entity object in behavior, integrating the individual micro behavior with the overall attribute. In this research, the AEM is applied to describe the individual behavior of the crowd, including the behaviors of following, detour, obstacle avoidance, etc., which can clearly illustrate the characteristics of crowd evacuation.

## 2. Theoretical Framework

The agent was originally derived from the research on distributed artificial intelligence launched by the Massachusetts Institute of Technology in the 1970s. It was an individual with intelligence and independent behavior, which can coordinate with each other according to the state reflected by other individuals and the system in order to determine its behavior and state [31]. Drawing on this idea, the model in this research defines pedestrians as individuals with decision-making abilities and unique attributes who can make real-time decisions according to dynamic and static information in the evacuation environment in order to determine their own evacuation behavior. See Figure 1.

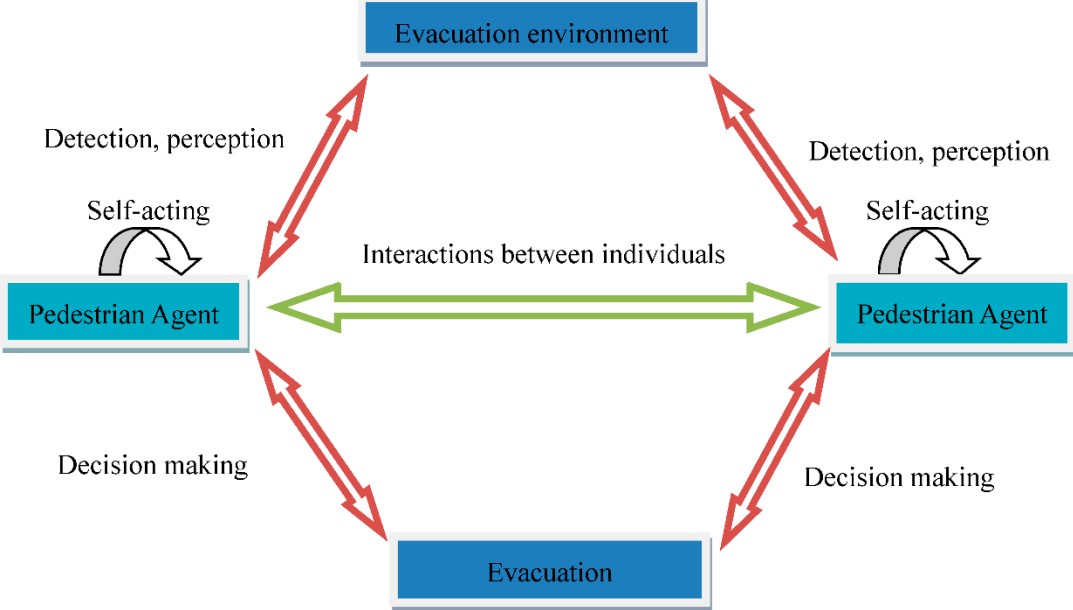

**Figure 1.** Theoretical model of the AEM.

The AEM in this paper has three characteristics: autonomy, reactivity, and sociality. Autonomy is the essential characteristic of an agent which can perceive the state of its environment and its position, and its behavior is driven by the goal of safe evacuation. The AEM can control its internal state and behavior without external intervention, including the maximum walking speed, the minimum reaction time, behavioral decision-making characteristics, and preferences.

Reactivity refers to the ability that an agent determines its impact on evacuation, and responds promptly to adapt to changes in the evacuation environment by sensing the environment. The AEM sets up a detection system for the evacuation agent, through which the system senses whether there are other pedestrians or obstacles within a certain range; it then makes behavioral decisions based on different detection results, eventually taking responses to follow or avoid pedestrians or obstacles. Through this mechanism, the mechanical limitations of the social force model can be avoided to some extent. As shown in Figure 2, the hollow circle represents the moving individually, the arrow represents the direction of its movement, and the black solid circle represents other persons or obstacles. According to the social force model, the repulsive force generated by the left side of the travel direction is greater than the repulsive force generated by the right side, so the individual needs to make a detour to the right side, as shown in the solid line on the right side. However, in reality, most people follow the shortest path and choose the path shown on the dotted line. To avoid unnecessary behavior, the AEM makes optimal decisions and takes appropriate actions by sensing the environment so that most pedestrians can follow the optimal path in the process of evacuation.

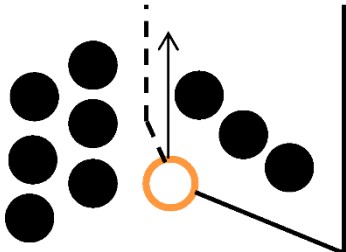

**Figure 2.** Unnecessary path developed by the social force model.

Sociality means that the evacuees do not exist independently, and they exchange information and coordinate actions with other agents at the evacuation scene. Multiple agents can also form an evacuation group.

According to the above characteristics of the agent, this paper establishes an agent-based evacuation model based on the kinematics model, the core component of which is a detection algorithm and behavioral decision making used to imitate pedestrians' walking decision-making and motion patterns.

## 3. Basic Algorithm of the AEM

The general behavior process of pedestrians in the evacuation process can be concluded as follows: first, determining the general direction of evacuation and perceiving the surrounding environment visually or in other ways when walking. When other pedestrians or obstacles are found, predictions are made based on whether there will be a collision according to their speed; the countermeasures are decided, such as detouring and following; and then the first step of the general direction decision is returned to. Therefore, the AEM is also carried out according to the above process in the evacuation process: (1) firstly, a certain detection range is divided according to its motion characteristics and the surrounding environment is detected; (2) the agent detects the obstacle and judges the attributes of the obstacle. After it enters the reaction zone set up to avoid the collision, the agent makes a behavioral decision and invokes the corresponding algorithm to behave as detouring or following. If the agent successfully avoids obstacles, it will return to the first step and move forward. If obstacle avoidance is not successful, the model continues to call the relevant algorithm to avoid obstacles. The basic operation process is shown in Figure 3.

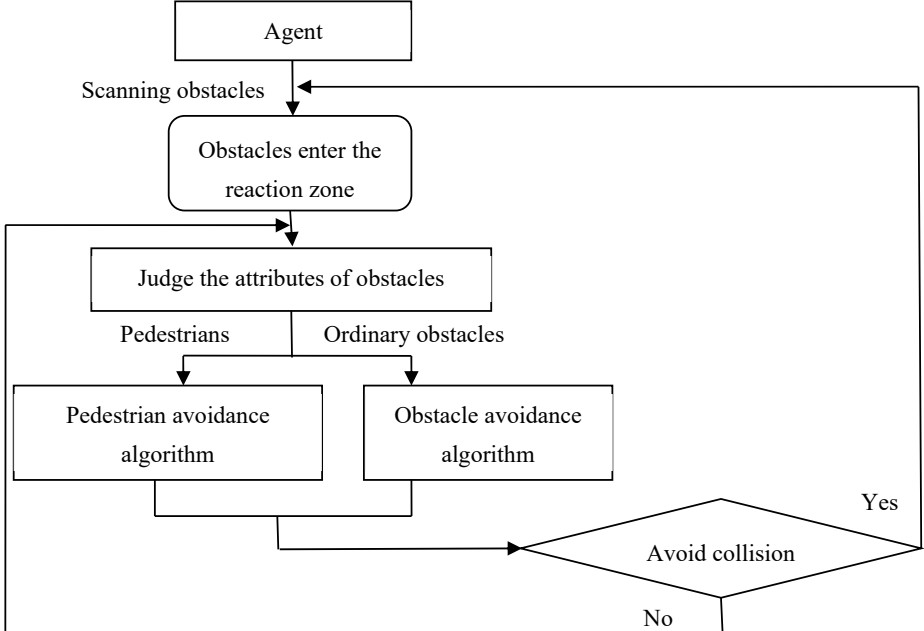

**Figure 3.** The flow chart of the agent walking.

### 3.1. Kinematics Model

The agent follows Newton's second law while walking, so the basic dynamics equation is:

$$m_\alpha \frac{d^2 \vec{n}}{dt^2} = \vec{F}_\alpha \tag{1}$$

where $m_\alpha$ is the mass of the pedestrian $\alpha$; $\vec{F}_\alpha$ is the resultant force of the pedestrian $\alpha$; and $\vec{n}$ is the vector formula of the moving direction.

It can be assumed that the current coordinate position of the pedestrian is $(x_m, y_m)$ and the desired target position is $(x_n, y_n)$. Then, $\vec{n}$ can be described as Equation (2):

$$\vec{n} = \frac{(x_m - x_n, y_m - y_n)}{\sqrt{(x_m - x_n)^2 + (y_m - y_n)^2}} \tag{2}$$

$$\vec{F}_\alpha = \vec{F}_\alpha^{\,driving}(\vec{v}_\alpha, v_\alpha^l \vec{e}_\alpha) + \vec{F}_{\alpha\beta}^{\,repulsion}(\vartheta_\beta(v_\alpha), \vec{e}_\alpha, L_{\alpha\beta}) + \vec{F}_{\alpha\beta B}^{\,steering}(\vartheta_\beta(v_\alpha), \sum_B F_{\beta B}(\varphi)) \tag{3}$$

where $\vec{F}_\alpha^{\,driving}$ is the driving force that pedestrians are subjected to, $\vec{F}_{\alpha\beta}^{\,repulsion}$ is the repulsive force on the pedestrian $\alpha$, and $\vec{F}_{\alpha\beta B}^{\,steering}$ is the steering force that the pedestrian $\alpha$ receives. $\vartheta_\beta(v_\alpha)$ is a function of selecting the obstacle $\beta$, which is the obstacle requiring priority processing in its detection range. $\sum_B F_{\beta B}(\varphi)$ is the direction of steering, $B$ is the set of all obstacles within the detection radius, and $F_{\beta B}$ represents the positional relationship of other obstacles in $B$ with $\beta$. Equations (4)–(6) below provide further descriptions of the parameter variables of Equation (3).

The driving force is determined as Equation (4):

$$\vec{F}_\alpha^{\,driving}(\vec{v}_\alpha, v_\alpha^l \vec{e}_\alpha) = \frac{m_\alpha \times (v_\alpha^l \vec{e}_\alpha - \vec{v}_\alpha)}{\tau_1} \tag{4}$$

where $v_\alpha^l$ is the maximum speed; $\vec{e}_\alpha$ is the desired direction; $\vec{v}_\alpha$ represents the actual speed; $v_\alpha^l \vec{e}_\alpha$ is the desired speed; and $\tau_1$ represents the response time.

Personnel in the process of constantly walking by the exit of the attraction drive to ensure that they do not deviate from the correct direction of travel. At the same time, different positions of personnel to exert the attraction of the point location are also different. As shown in Figure 4a, the area is divided into two parts, and the coordinates of the points corresponding to the attraction forces are determined differently in different parts. For the person in area 1, the location of the attraction applied by the exit is the forward projection point on the exit, and for the person in area 2, it is the attraction point on his or her side. The endpoints of the attractive force range on the side where they are located, as shown in Figure 4b.

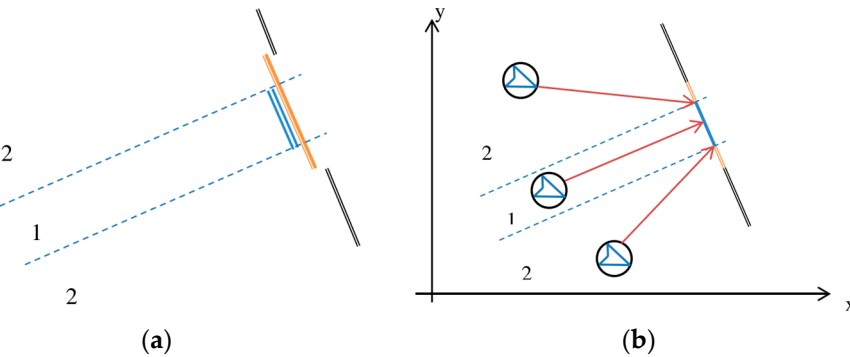

(**a**)　　　　　　　　　　　　　　　　　（**b**）

**Figure 4.** *Cont.*

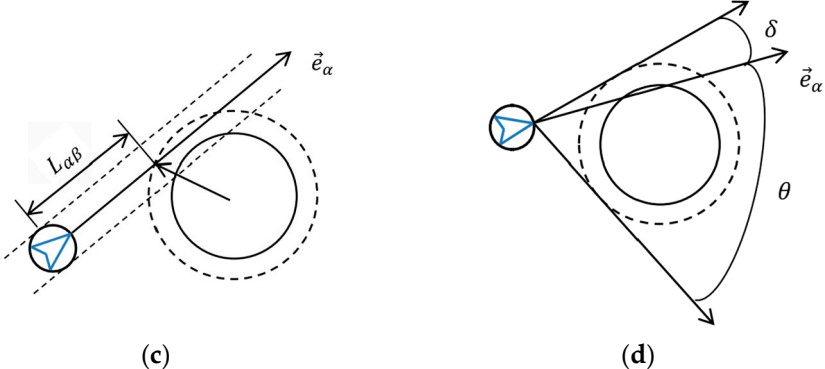

**Figure 4.** Schematic diagram of force. (**a**) Human location according to export; (**b**) export attractiveness; (**c**) application of reject force; (**d**) angle of deflection.

The endpoints of the attractive force range on the side where they are located, as shown in Figure 4b. The repulsive force is applied to prevent collisions, and the magnitude of the applied force is related to the position of the pedestrian $\alpha$ with respect to the blocking object $\beta$ relative to their position. As shown in Figure 4c, the radius of the circular obstacle plus the concentric dashed circle of the radius of the personnel The direction of travel of the personnel is $\vec{e}_\alpha$; moreover, in order to avoid collision with the obstacle without turning to avoid the premise, it is necessary to decelerate to zero within a distance $L_{\alpha\beta}$. The repulsive force can be described as Equation (5).

$$\vec{F}_{\alpha\beta}^{\text{repulsion}}(\vartheta_\beta(\vec{v}_\alpha), \vec{e}_\alpha, L_{\alpha\beta}, t) = -\left\{ \frac{2L_{\alpha\beta} m_\alpha}{(\tau_1 + \tau_2)^2} - \frac{\vec{F}_\alpha^{\text{driving}} \cdot \vec{e}_\alpha}{|\vec{e}_\alpha|} \right\} \cdot \vec{e}_\alpha \tag{5}$$

where $\vec{v}_\alpha$ represents the current speed. $L_{\alpha\beta}$ represents the minimum distance in which a pedestrian is an agent to an obstacle element in the direction of moving. $\tau_2$ represents the reaction time of the pedestrian. $\frac{\vec{F}_\alpha^{\text{driving}} \cdot \vec{e}_\alpha}{|\vec{e}_\alpha|}$ is the component of $\vec{F}_\alpha^{\text{driving}}(\vec{v}_\alpha, v_\alpha^l \vec{e}_\alpha, t)$ in the $\vec{e}_\alpha$ direction, and subtracting this can prevent the repulsive force from being offset by the component of the driving force in that direction. The steering force is shown in Equation (6):

$$\vec{F}_{\alpha\beta B}^{\text{steering}}\left(\vartheta_\beta(v_\alpha), \sum_B F_{\beta B}(\varphi), t\right) = \vec{F}_\alpha^{\text{direction}} \tag{6}$$

$\vec{F}_\alpha^{\text{direction}} = 2m_\alpha v_\alpha^l \vec{e}_\alpha A$ is the force applied in the direction of moving to avoid the obstacle, which can be guaranteed to accelerate to the maximum speed during the response time. A is a two-dimensional matrix, representing a rotation of $\vec{e}_\alpha$ counterclockwise by a $\delta$ angle. As shown in Figure 4d for the pedestrian $\alpha$, there are two directions to avoid the obstacle by steering choose, and the specific one can be determined by the function $\sum_B F_{\beta B}(\varphi)$, and choosing different directions corresponds to different deflection angles and different deflection forces. The rotation matrix is shown in Equation (7):

$$\begin{bmatrix} \cos\delta & \sin\delta \\ -\sin\delta & \cos\delta \end{bmatrix} \tag{7}$$

The steering force does not always exist when the agent walks, and it will be active according to the behavior of the detection result of the surrounding environment. When an agent detects an obstacle or other agent, it will make a behavior decision to bypass, wait, or follow. These behaviors have a lot to do with the personality characteristics of pedestrians. The model assigns the agent-to-personality features through a random number generation function (Equation (8)), and the generation of random numbers conforms to the

standard normal distribution. When the character of the evacuation person is initialized, the individual will be set to a mild or aggressive attribute according to the numerical value of the random number.

$$f(u) = \frac{1}{2\pi} e^{-\frac{u^2}{2}} \tag{8}$$

### 3.2. Detection Algorithm

In the AEM, a whole circle is used to represent an evacuated individual, in which the diameter of the circle is determined by the length of the shoulder width of the individual. Using 40 cm as the default diameter for pedestrians in this model, the individual diameter values are not fixed and can be adjusted according to needs.

Under normal circumstances, pedestrians perceive the surrounding environment through vision. Safety ergonomics state [32] that when a human being is stationary, the binocular viewing area will be approximately 60° or so. Pedestrians can expand their viewing angles by slightly deflecting their heads during walking. Therefore, it is possible to set the agent-based to obtain a field of view of 90° on the left and right sides only when the body is not rotated; with this in mind, in the model, the scanning detection angle of the agent in the forward direction is set to 180°.

The setting of the detection range of the agent-based is mainly considered from the viewpoint of avoiding a collision. Under normal circumstances, people need to consider the safety distance with other people and obstacles while walking, especially when the speed is faster and the safety distance is shorter, but the collision will occur because there is no time to slow down or turn. The pedestrian's safety distance is determined by its speed and reaction time. In traffic engineering, it is described that the driver needs a minimum of 0.4 s of sensation–reaction time before starting the braking. The braking effect is 0.3 s and the total braking process is 0.7 s. In this paper, some parameters in Pathfinder are used to calculate the safe distance. Pathfinder's technical documentation gives $a_{max} = 2 * v_{max}$; in other words, it takes 0.5 s to slow down from the maximum speed to zero, so the total reaction time plus the response time is about 0.9 s. The reaction distance and response distance corresponding to the reaction time and the response time are both shown in Figure 5.

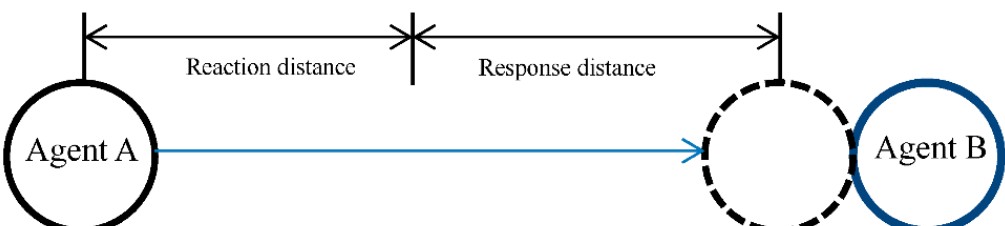

**Figure 5.** Reaction and response distance.

### 3.2.1. Detection Range

Based on the above ideas, a detection model of the AEM is established (see Figure 6). The circle in the figure represents the agent, and the internal arrow is its current direction of motion. Arrows $r_1$ and $r_2$ represent the detection distance along the current direction of motion, the length of which varies with speed. Arrows 2 and 3 mainly detect lateral obstacles, and the length can be set to a fixed value.

At each time step of the program running, the agent determines whether there is an obstacle in the way by sensing the surroundings within the detection range and then formulating the subsequent measures. The detection range of the evacuation agent is the semi-circular arc whose radius is the radius of the detection center, as mentioned before, while the horizontal field of view is about 180° when the person is moving forward without rotating the body and only slightly deflecting the head; simultaneously, it also simplifies the calculation.

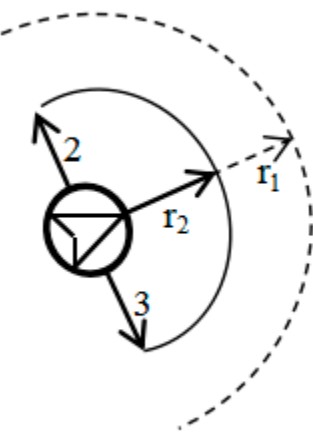

**Figure 6.** Detection model.

As shown in Figure 6, the actual detection radius depends on two factors. One is the dynamic detection radius $r_1$ determined by its oelocity, which varies with the change in velocity; the other is the fixed ength detection radiu $r_2$, which is related to visibility.

3.2.2. Obstacle Detection

The purpose of obstacle detection is to determine the obstacles that the agent needs to prioritize. Secondly, through the detection process, the distribution of obstacles in the direction of advancement is summarized and used as the basis for making decisions based on subsequent walking behavior.

In order to determine whether there are obstacles in the detection area, two sets of coordinate systems are established (one is the global coordinate system XY and the other is the local coordinate system st). The XY global coordinate system is transformed into the st local coordinate system by rotating angle θ, where the s axis is the forward direction of the agent. The point P (x, y) in the original coordinate system is transformed into (s, t) in the new coordinate system after the transformation.

When detecting obstacles, this research simplifies all obstacles (except walls) into circles to reduce computational difficulties. Therefore, the question of whether other pedestrians or objects are within the detection radius is linked to figuring out the distance between pedestrians or pedestrian and objects, i.e., the distance between the circles or the circle and the line segment.

It can be assumed that the center coordinates of two circles are $A(x_{r1}, y_{r1})$ and $B(x_{r2}, y_{r2})$ with radii $r_A$ and $r_B$, respectively. The distance $L_{A \to B}$ between the two circles is as follows:

$$L_{A \to B} = \sqrt{(x_{r1} - x_{r2})^2 + (y_{r1} - y_{r2})^2} - r_A - r_B \tag{9}$$

If the center point of an agent is $S(x_s, y_s)$, the starting point of the line segment of the wall is $M(x_m, y_m)$ and the end point is $N(x_n, y_n)$. Then, the vertical distance from point S to the wall is:

$$L_{s \to mn} = \frac{|Ax_s + By_s + C|}{\sqrt{A^2 + B^2}} \tag{10}$$

The distance between two circles and the distance from the point to the line are shown in Figure 7a, The individual $A_\alpha$ is stored in the set $C_i^{r1}$ by the detection radius of $r1$, and the contents that meet the conditions are stored in the set $C_i^{r1}$. $i$ represents the number of elements in the set, and the elements in the set need to be conditionally filtered, and the walls are not included in the set. The satisfied conditions contain the following.

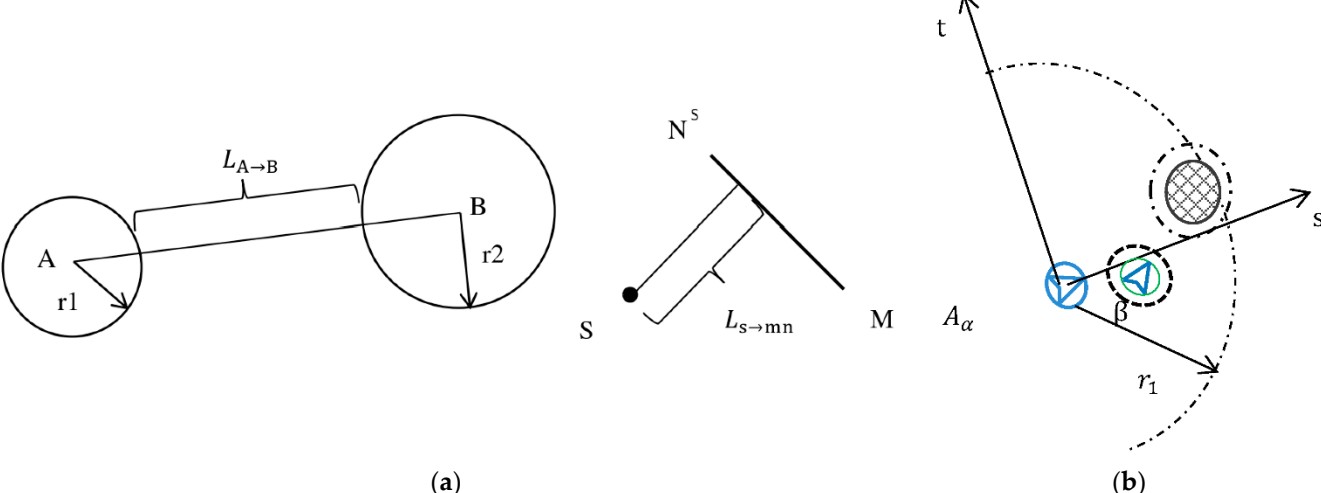

**Figure 7.** (**a**) Distance between two circles and point to the straight line. (**b**) Schematic diagram of the determination of the nearest obstacle element.

1. The obstacles and pedestrians detected by the individual $A_\alpha$ are stored in the set $C_i^{r_1}$, where $I$ represents the number of obstacles and pedestrians in the set;

2. If the obstacle element $C_i^{r_1}$ satisfies the s-axis coordinate greater than 0 in the st local coordinate system, it is stored in the set $S_j^{r_1}$, where $j$ represents the number of elements in the set $j \leq i$;

3. The obstacles that have a direct influence on the direction of travel of $A_\alpha$ in the set $S_j^{r_1}$ are filtered out, i.e., the absolute value of the t-axis coordinates of the element in the local coordinate system is smaller than the radius of its own plus the radius of $A_\alpha$, and is stored in the set $N_k^{r_1}, k \leq j$;

4. According to the above algorithm, the obstacle element closest to $A_\alpha$ in the set $N_k^{r_1}$ is determined and labeled as β. Figure 7b shows the calculation schematic diagram, and the individual in the green border is the nearest individual to $A_\alpha$.

*3.3. Behavioral Decision Making*

In the model, the moving direction is decided by the environment and the evacuees' characteristics instead of being driven by force. For example, if a person moves in the same direction in front of the pedestrian, the social force model will generally exert a force on the pedestrian to avoid the person in front, which makes it difficult to flexibly simulate the following, waiting, or surpassing behaviors. However, in reality, pedestrians will make different decisions according to the characteristics of the evacuation environment and their conditions after encountering obstacles. Some prefer to follow or wait, some prefer to go beyond, and some choose the direction with a lower personnel density. Therefore, the AEM presented in this research is a combination of the social force model and the decision-making model to simulate evacuation behavior more realistically.

In the AEM, after the agent detects the nearest obstacle element, it will adopt the strategy of following, circumventing, or transcending. As for the direction of circumventing or transcending, it will take the density of personnel within a certain range as the decision basis. The agent will decide the direction of the next step according to the density distribution of obstacles in the front and the weight of the position, reflecting more intelligent pedestrian decision making. The calculation of the obstacles' density aims to calculate the distance between the obstacles in the set $S_j^{r_1}$ and the agent $A_\alpha$ except for the nearest pedestrian β, as well as determine whether it is located above or below the x-axis in the local coordinate system. The qualified obstacles can be seen in Table 1.

**Table 1.** Pedestrian location statistics for behavioral decision making (1).

| Position | $D_1 \geq D_2$ | $D_1 < D_2$ |
|---|---|---|
| Above the s-axis | $S_1$ | $S_2$ |
| Below the s-axis | $S_3$ | $S_4$ |

In Table 1, $S_1$, $S_2$, $S_3$, and $S_4$ are integers that represent the number of people, and the sum value of them should be less than or equal to $i - 1$. $D_1$ is the distance between the agent $A_\alpha$ and another obstacle, and $D_2$ is the sum of the radii of the two agents + the diameter of $A_\alpha$.

In addition to reacting to obstacles in the range of $r_1$, the agent $A_\alpha$ may also choose the direction of detour according to the distribution density of obstacles in the field of vision. Therefore, besides counting the number of people in the detection range of radius $r_1$, it is necessary to detect the range between $r_1$ and visibility $r_2$. However, since only the direction of the detour is judged, it is not necessary to detect the front side. Therefore, it is only necessary to perform the detection and statistics on the shaded portion in Figure 8, and the result is shown in Table 2.

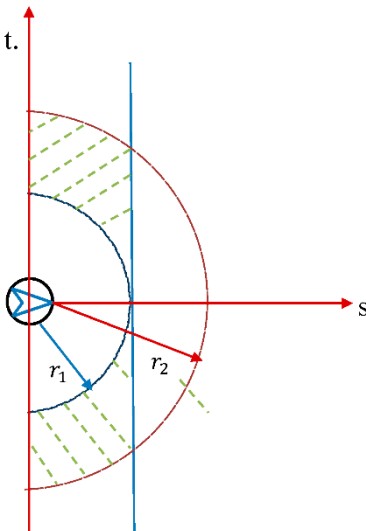

**Figure 8.** Determination of the statistical scope.

**Table 2.** Pedestrian location statistics for behavioral decision making (2).

| Position | Above the s-Axis | Below the s-Axis |
|---|---|---|
| Number of people | $S_5$ | $S_6$ |

The statistical information can be summarized and different weights can be assigned to the number of different locations, as shown below:

$$S_{up} = \mu S_1 + \sigma S_2 + \epsilon S_5 \tag{11}$$

$$S_{down} = \mu S_3 + \sigma S_4 + \epsilon S_6 \tag{12}$$

$\mu$, $\sigma$, and $\epsilon$ are the different weight values, respectively, where $\mu > \sigma > \epsilon$ and $\mu + \sigma + \epsilon = 1$.

By comparing the values of $S_{up}$ and $S_{down}$, the density of people around the agent $A_\alpha$ can be judged. If the values of $S_{up}$ and $S_{down}$ are both large, it can be shown that the pedestrian density in the direction of the agent moving forward will be relatively large, and there will be little space to bypass within the visible range, so the agent $A_\alpha$ will choose to follow the forward agent to move on.

When the values of $S_{up}$ and $S_{down}$ are both small, which means that there is a small density of people around the agent $A_\alpha$, there will be enough space to bypass or surpass,

but the choice to bypass, surpass, or keep following depends on the personality attribute of the agent. In this case, the probability $\varphi$ is introduced to assist evacuation decision making. When initializing evacuation attributes, agents are set to mild or radical attributes according to the values of random numbers.

3.3.1. Algorithm for Avoiding Pedestrians

After the agent detects the pedestrian in front, it will adopt a strategy of following or bypassing. This section discusses the algorithms of these two behaviors in detail.

Algorithm for Following

When agent A needs to follow agent B in front, he is required to slow down and maintain a safe distance from agent B, and it is necessary to apply a repulsive force to agent A. In the model, the magnitude of the force is related to the speed and the distance between the two, similar to the elastic force of the spring.

Under the action of this repulsive force, the speed of agent A decreases to the same as that of agent B in front, and agent A maintains this speed and keeps a safe distance from agent B. When agent B no longer blocks agent A, agent A returns to its original speed immediately. The application of this repulsive force is described in more detail in Section 3.1.

Algorithm for Bypassing

The first step in the detour algorithm is to perform collision detection on the front pedestrian. At this time, the center of agent B can be seen as the center and the radius is the sum of the radii of A and B which are used to draw a concentric circle of B to help determine whether the two will collide. When the ray along the direction of agent A has no intersection with the concentric circle of B, it means that A will not have any contact with B if it walks in the current direction. When the direction ray of A and the concentric circle of B only have one intersection point, it means that they will be tangent if they continue to move in the current direction. When the direction ray of A has two intersection points with the concentric circle of B, A will collide with B when walking in the current direction, and the location of the collision point will be closest to the current position of A among the two intersection points, as shown in Figure 9. When the latter two situations occur, agent A needs to detour.

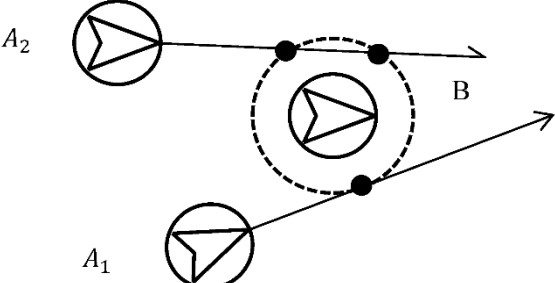

**Figure 9.** Sketch of collision detection.

Next, when agent A decides to bypass B, it is necessary to determine the angle of deflection. The coordinates of A set in the local coordinate system are $(x_A, y_A)$, the coordinates of B in the local coordinate system are $(x_B, y_B)$, and the distance between A and B is $L_{A \rightarrow B}$:

$$L_{A \rightarrow B} = \sqrt{(x_A - x_B)^2 + (y_A - y_B)^2} \tag{13}$$

As shown in Figure 10, assuming that the radius of B and A are $r_{A_\beta}$ and $r_{A_\alpha}$, $\theta$ and $\theta_1$ can be determined as:

$$\theta = \sin^{-1} \frac{r_{A_\beta} + r_{A_\alpha}}{L_{A_\alpha \to A_\beta}} = \sin^{-1} \frac{r_{A_\beta} + r_{A_\alpha}}{\sqrt{(x_\alpha - x_\beta)^2 + (y_\alpha - y_\beta)^2}} \tag{14}$$

$$\theta_1 = \tan^{-1} \frac{\left| t_{A_\beta} \right|}{s_{A_\beta}} \tag{15}$$

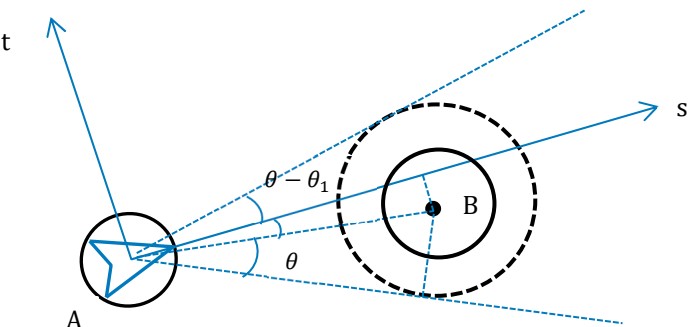

**Figure 10.** Angle of deflection.

When A decides to deflect in the positive direction of the t-axis, the angle of deflection is $\theta - \theta_1$, and when deflect in the negative direction of the t-axis, the angle of deflection is $\theta + \theta_1$.

### 3.3.2. Algorithm for Obstacle Avoidance

Except for other pedestrians, when the agent detects obstacles or walls, it will also be repulsive. Taking the wall as an example, the direction of force is along the normal direction of the wall, and the size is the same as the repulsive force in the dynamic model. The force is used to balance the force exerted by the outlet in the normal direction of the wall. The trajectory of the final movement is shown by the dotted line in Figure 11.

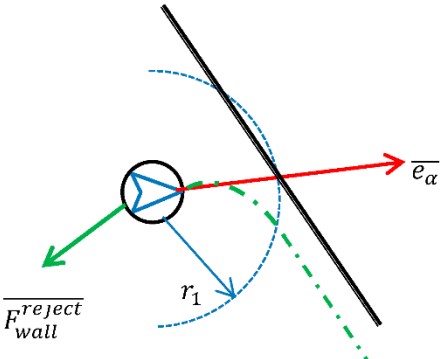

**Figure 11.** Repulsive force of a wall.

In the algorithm, the boundary of the obstacles perpendicular to the direction of the travel of personnel is replaced by some small circle to simplify the geometric calculation. When the agent detects an obstacle, it will perform obstacle avoidance calculations for different small round obstacles. The required deflection direction of each small circle is represented by arrows of different colors (see Figure 11). The agent is deflected in order from these angles until the obstacle is bypassed. The theoretically derived trajectory is shown by the thick blue line in Figure 12.

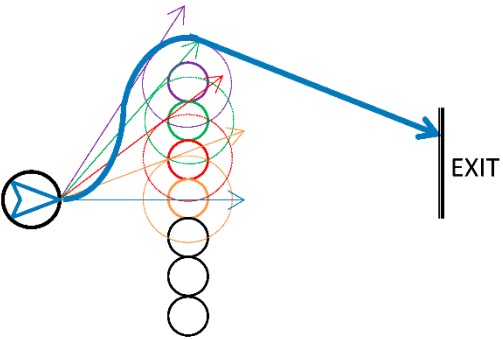

**Figure 12.** Schematic diagram of avoiding obstacles.

In this section, the intelligent decision model theory based on Newton's second law of motion is established, and the part of applied force includes the driving force, the repulsive force, and the steering force, and the decision model and the probability model are introduced into the mechanics model compared with the traditional social force model, resulting in a new evacuation model. Intelligent behavior is mainly reflected in two aspects: speed control and direction selection. The speed control of an individual is mainly influenced by the density of people around itself, and the truncated operation of the maximum speed of an individual is implemented based on the experimental results and data from other research scholars. The choice of direction includes obstacle avoidance and the choice of optimal exit. Obstacle avoidance includes avoiding obstacles, other people, and walls, and the choice of optimal exit is determined by two factors: the distance from the individual to the exit and the number of people to be evacuated from that exit. The decision model is mainly embodied in the steering force; it needs to first determine the detection range of the first impact on its own direction of travel of the individual, and then determine the detection range of other individuals, obstacles, and the individual's location information, so as to determine the direction of avoidance. Regarding the probability model embodied in the evacuation of individuals in the direction of travel when they encounter another movement of people, or when choose to follow or go around, the probability value used is the standard normal distribution that generates random values of the standard normal-terms distribution, and determines the behavioral attributes of different individuals based on the random numbers.

## 4. Simulation Examples

### 4.1. Simulation of Walking Behavior to Avoid Pedestrians

This section uses the above algorithm to simulate the common behaviors in the evacuation process, including following, bypassing, and going in opposite directions.

#### 4.1.1. Simulation of Following

In the simulated scene, agents A and B are on the same horizontal line and move toward the exit at the same time. The two agents are 3 m apart and A is 10 m away from the exit. The initial speed of A is 0 m/s and will reach 1.5 m/s after it starts to walk. Speed B always stays walking at a speed of 1 m/s (see Figure 13).

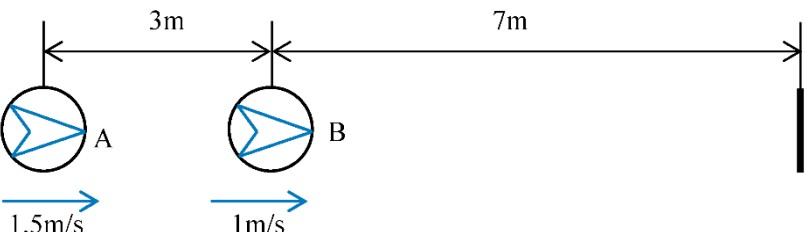

**Figure 13.** Scene of the following behavior.

The program running results are shown in Figure 14. The speed of agent A has a process from starting stationary to accelerating to the maximum. As the radius of the detection range set by the algorithm is proportional to the speed, the size of the circular box representing the detection range of agent A in the figure is constantly changing.

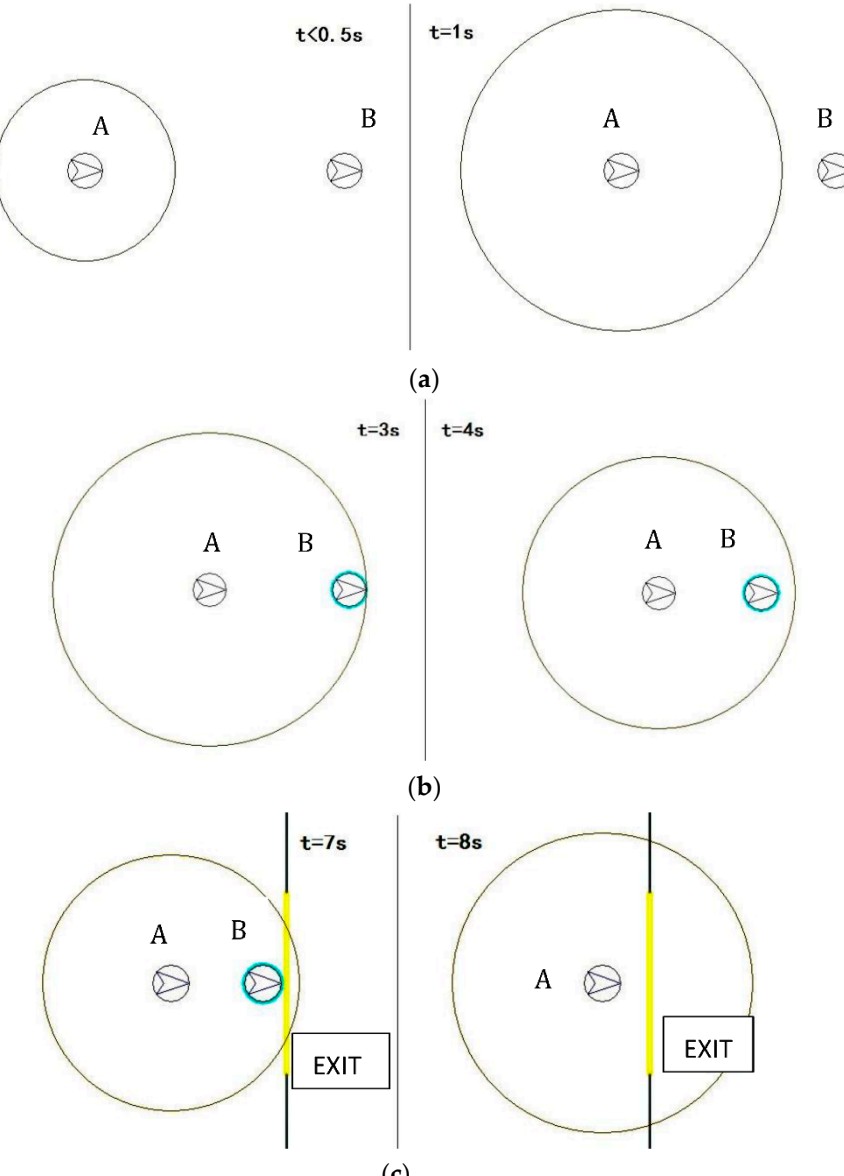

**Figure 14.** (**a**) Walking state at 0.5 and 1 s. (**b**) Walking state at 3 and 4 s. (**c**) Walking state at 7 and 8 s.

Since there has been a speed difference between the two agents, the front agent B enters the detection range of agent A at the third second, and the program marks B with a blue frame automatically. In the example, agent A chooses to follow the rear of agent B, so it is decelerated by the repulsive force, and then A and B will maintain a safe distance (see Figure 14b).

As shown in Figure 14c, at 7 s, agent B reaches the exit while agent A accelerates to approach the exit as there is no obstruction in front.

Figure 15 shows the speed-changing process of agent A and its speed comparison with agent B in the following behavior simulation. While walking, agent B is in the front and the speed of B remains unchanged at 1 m/s; in the meantime, the speed change in agent A can be divided into three stages.

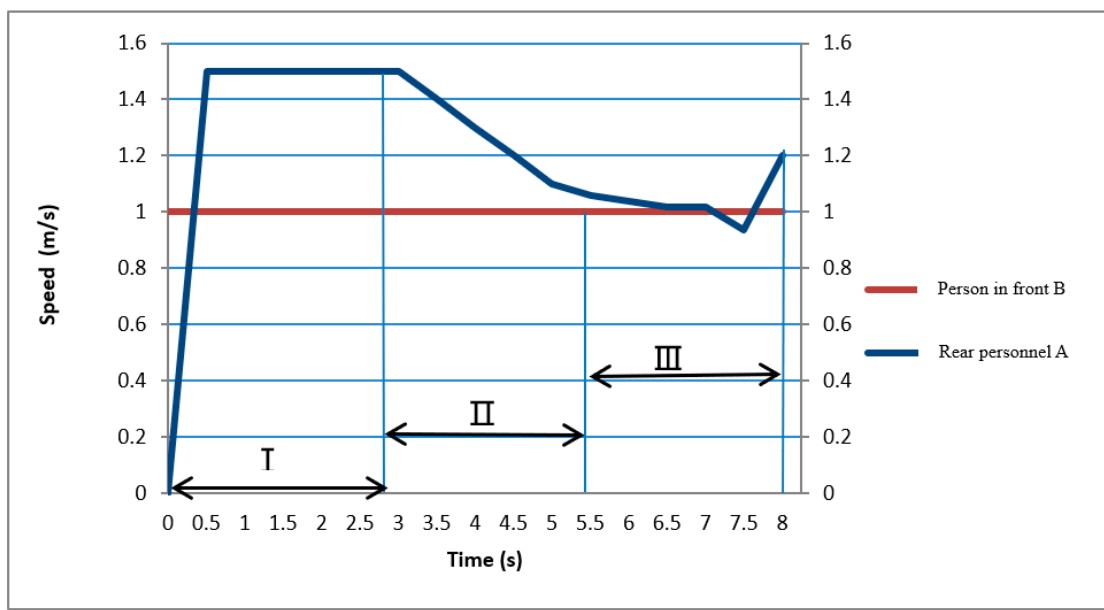

**Figure 15.** Speed change process of agent A.

In the stage, agent A accelerates from 0 through 0.5 s to a maximum speed of 1.5 m/s and then moves at a steady speed of 1.5 m/s.

In stage II, at about 3 s, agent B enters the detection range of agent A, and A decides to follow B. After 2.5 s, agent A decelerates to 1 m/s and maintains a safe distance from B at a constant speed.

In stage III, at 7 s, B reaches the exit. At this time, A starts to increase the speed because there is no obstacle, and then quickly reaches the exit, and the speed is about 1.2 m/s at that time.

This example demonstrates the complete dynamics of the following behavior of agent A from the acceleration of the pedestrian to the normal speed while the front pedestrian B has a steady speed, including the adoption of deceleration to follow, and increases the speed after the de-detection of an obstacle, which can illustrate a high degree of simulation.

4.1.2. Simulation of Human Flow in the Opposite Direction

The evacuation scene is shown in Figure 16 containing four pedestrians on both sides of the corridor, running in the opposite direction without sequence. The pedestrian marks on the left side are 0, 1, 2, and 3, and the pedestrian numbers on the right side are 4, 5, 6, and 7. The exit is located on both sides of the corridor. The pedestrian speed is 1.5 m/s and the corridor length is 7 m.

t=0s

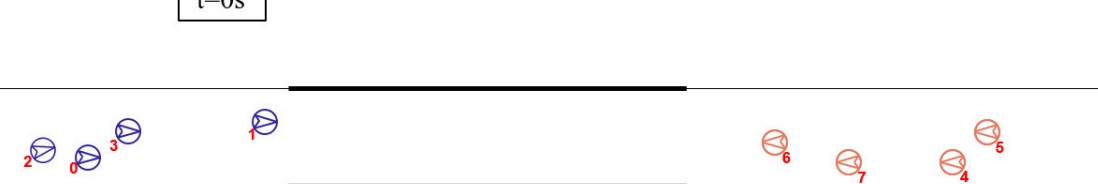

**Figure 16.** Evacuation scene setting of opposite pedestrian flow.

As shown in Figure 17a, when t = 4 s, pedestrian 1 on the left and pedestrian 6 on the right meet and turn. Pedestrian 1 and pedestrian 6 choose to turn to the left of their respective directions of advance to avoid the other side. After the success of avoidance, 1 and 6 continue to move forward. At this time, t = 5 s, as shown in Figure 17b. When t = 6 s,

pedestrian 1 successfully avoids the opposite of pedestrian 5 and reaches the exit, as shown in Figure 17c; when t = 8 s, pedestrian 1 leaves through the exit, as shown in Figure 17d.

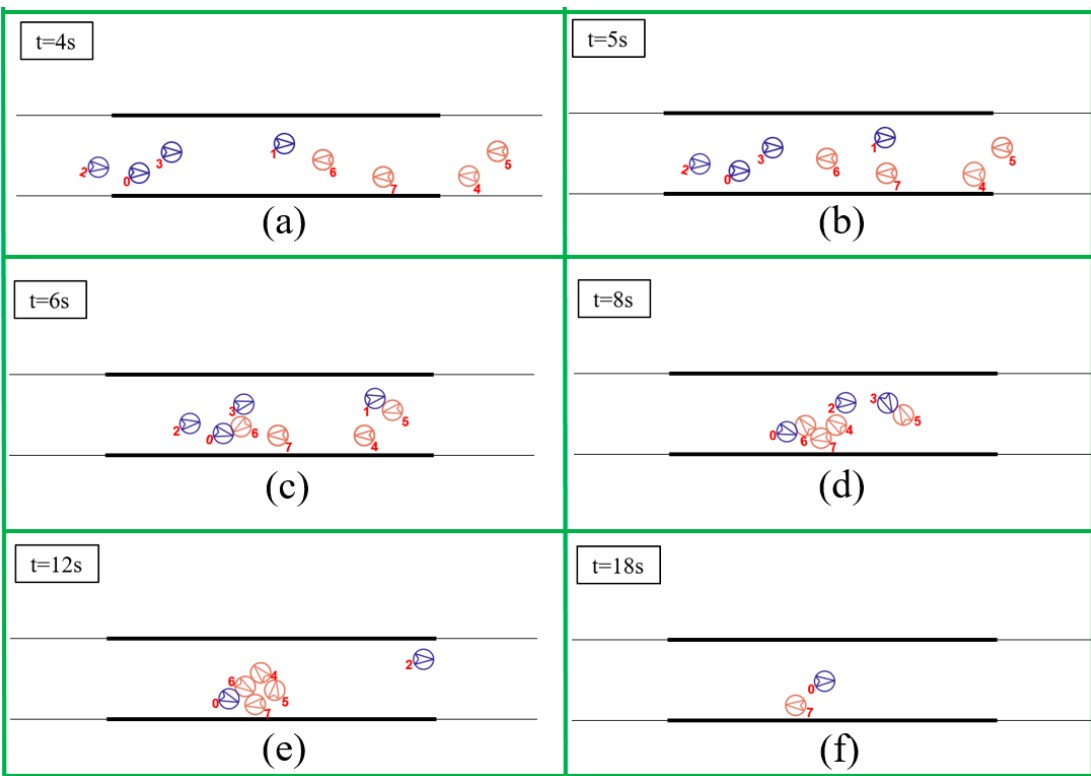

**Figure 17.** Screenshot of the simulation of human motion in the opposite direction.

As shown in Figure 17e, when t = 12 s, pedestrians 1, 2, and 3 on the left side reach the exit smoothly, and there is a traffic jam between the right pedestrian and zero pedestrians on the left side. Meanwhile, pedestrians 4 and 5 on the right side deflect to the right side of the forward direction, and choose to pass through the larger available space on their right side, before then continuing to move forward. Then, pedestrians 0 and 7 avoid each other successfully, and continue to move forward, as shown in Figure 17f, before then reaching the exit, at which point the evacuation is over.

The above example describes the opposite pedestrian flow behavior, simulating the evacuation process under the condition of multiple people moving in the opposite direction. The AEM better demonstrates the behavior of evacuees by continuously avoiding pedestrians in the process of moving forward. In the direction selection, evacuees choose the direction with a small density and a large available space, which effectively reflects the intelligent evacuation model.

In this example, pedestrians 1 and 6 in the opposite motion firstly show the decision-making process of avoiding pedestrians in the course of the opposite motion (enlarged details of pedestrians 1 and 6 in Figure 17a can be seen in Figure 18). Pedestrian 1 detects that the opposite pedestrian 6 is within the detection range, and pedestrian 6 also detects the opposite pedestrian 1 as well, and both of them make the decision to detour. When t = 4 s, both of them make a deflection trend, and the speed decreases to prepare for turning around. When t = 5 s, the two persons in opposite directions steer away from each other. After avoiding each other, the two continue to move till they reach the corresponding exit position. Figure 18c shows the route map of pedestrians 1 and 6. Figure 19 shows the track of pedestrians 0, 2, 3, 4, and 7 from t = 8 s to the end of the whole evacuation.

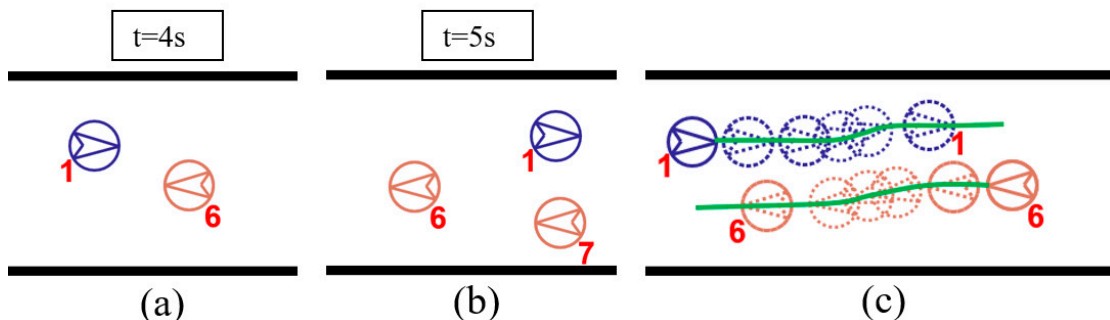

**Figure 18.** Screenshot of simulation of phase motion.

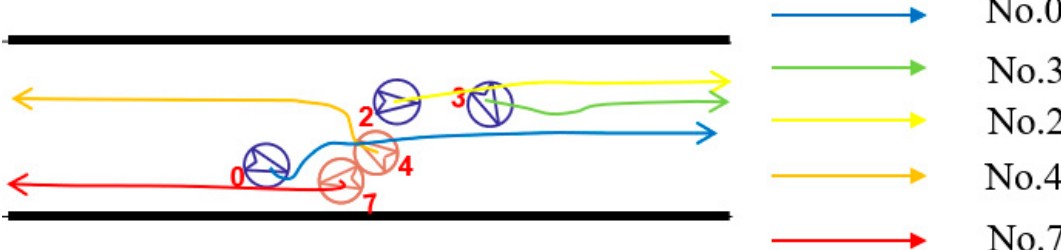

**Figure 19.** Schematic diagram of motion track.

### 4.2. Simulation of Avoiding Obstacles

For general obstacles, replacing the boundary of obstacles with a series of small circles can simplify the calculation of geometric relations and obtain a higher simulation effect. The following example selects an edge of an obstacle to demonstrate the effect of the obstacle avoidance algorithm proposed in this paper.

In the scene shown in Figure 20, one side of the obstacle is replaced by five circles with a radius of 0.4 m. Smaller circles are applied to obtain more accurate simulation results. The exit is located behind the obstacle, so the agent can only go up around the obstacle to reach the opposite exit.

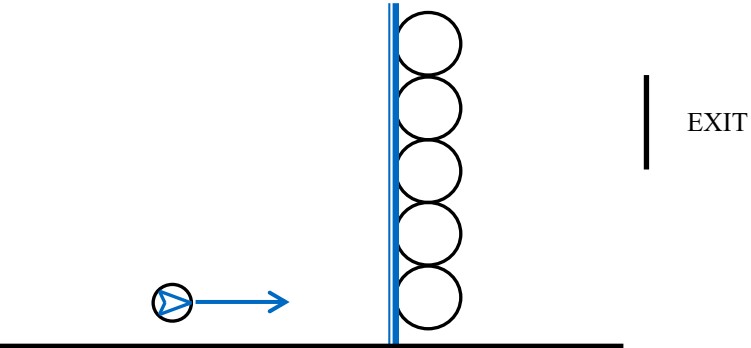

**Figure 20.** Schematic diagram of obstacle avoidance scene.

Since the boundaries of obstacles are replaced by a series of circles, the agent needs to constantly avoid each obstacle. The motion process of the agent is shown in Figure 21, where the yellow circles represent the obstacles that the agent needs to avoid at the current moment and the blue arrow represents the direction of the agent's rotation.

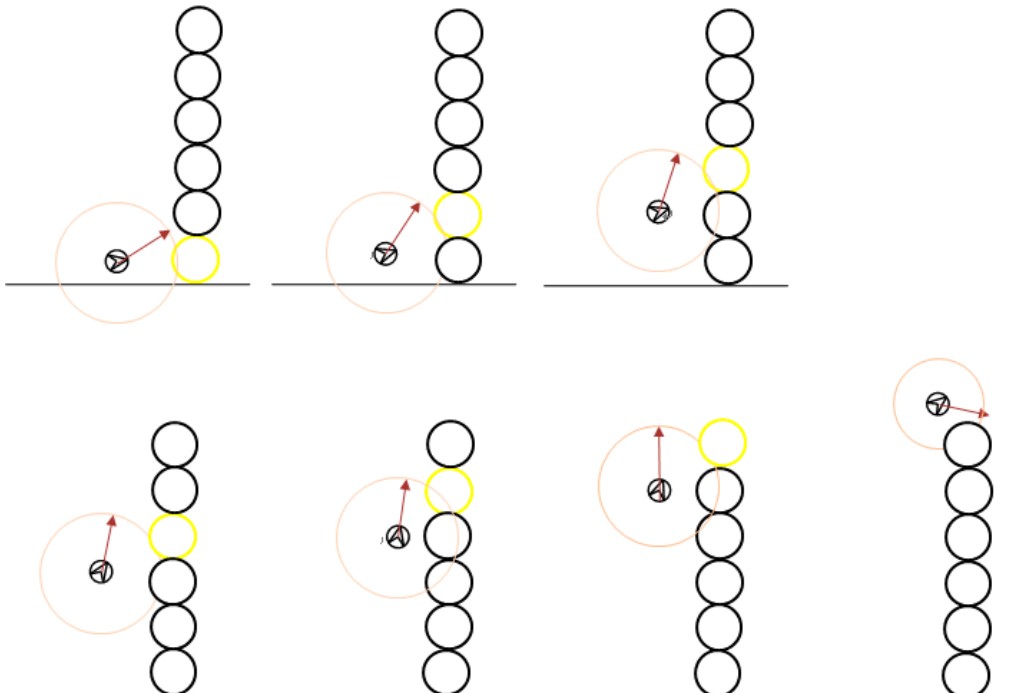

**Figure 21.** Schematic diagram of the agent avoiding obstacles.

*4.3. Simulation of Actual Scene Evacuation*

4.3.1. Simulation of Actual Scene

The total area of the convention and exhibition center is 7250 m², including an inactive area of 1681 m², and the area of the remaining personnel is 5569 m². There are 14 exits in the whole exhibition center, each with a width of 3.6 m. In this area, the number of random positions is 3000, and the personnel density reaches 0.54 person/m², as shown in Figure 22.

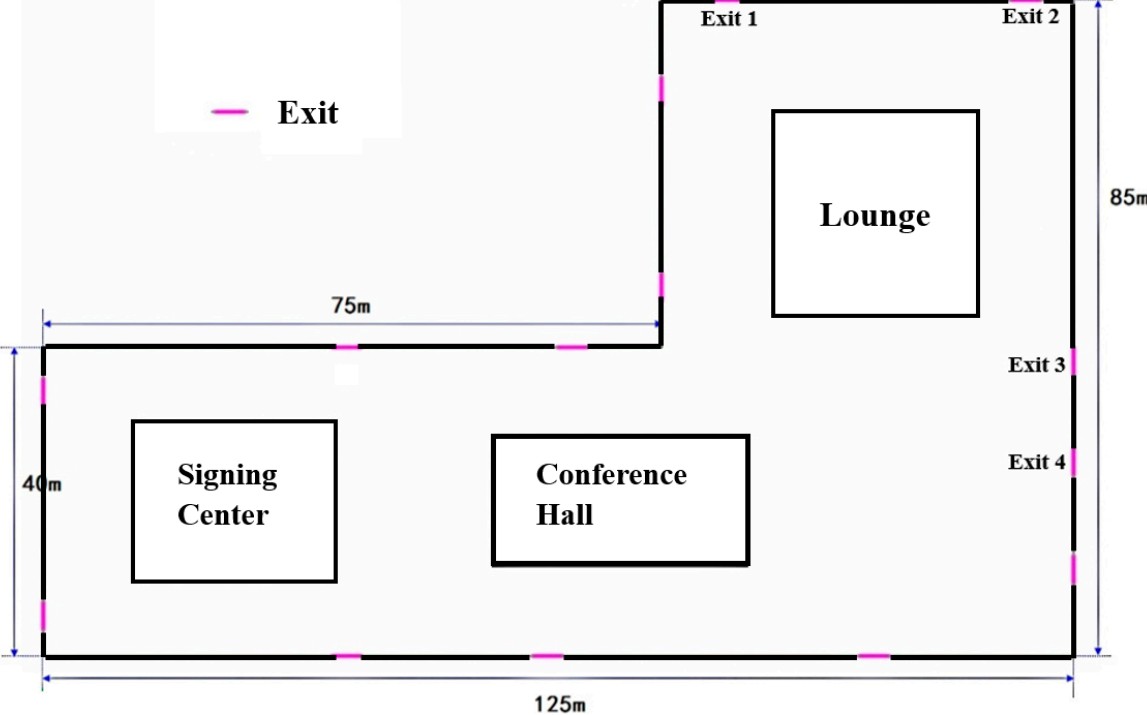

**Figure 22.** Exhibition center plan.

The evacuation process is shown in Figure 23. The model adopts the principle of the shortest path. When t = 2 s, evacuees begin to approach the corresponding nearest exit, and evacuees in the whole scene start to be divided into different areas. At the times of t = 4 s and t = 8 s, evacuees gather at the exit continuously, forming the arched distribution of evacuees.

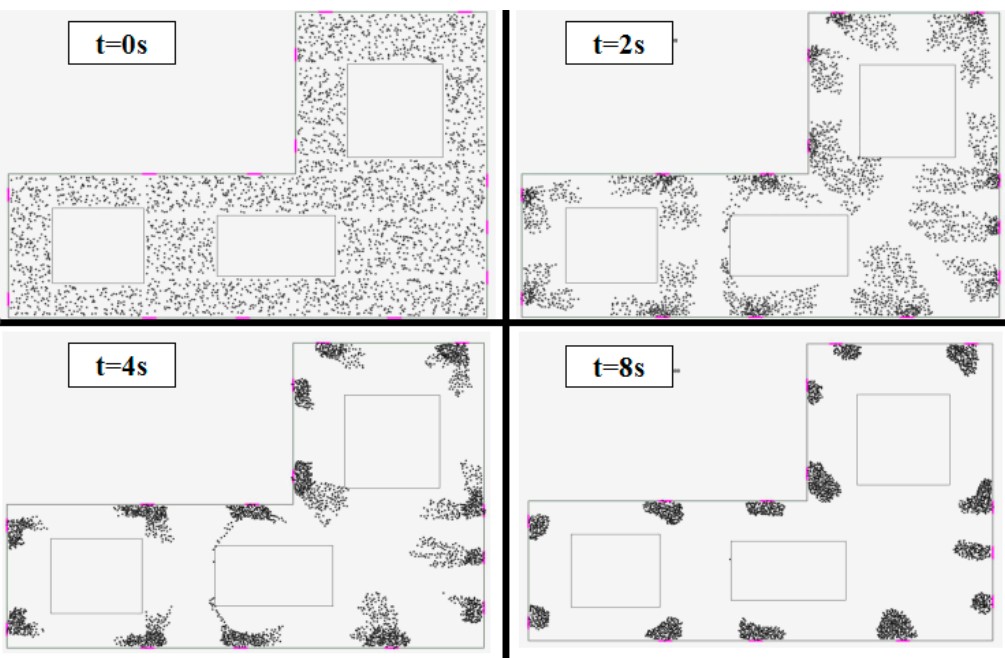

**Figure 23.** Screenshot of evacuation simulation of the exhibition center.

The whole evacuation is completed in t = 39.8 s. It can be seen from the evacuation process that the evacuation time is mainly consumed by waiting at the exit. Therefore, the evacuation time in this example depends on the number of exits. According to the code for the fire protection design of buildings, the article of gb50016-2014 (2018 Edition) shows that for an adult shoulder width of 0.55 m, the number of people passing through the 0.55 m-wide exits per minute is 43, and the width of per capita flow is 0.55 m. Therefore, each exit can form 3.6/0.55 = 6.5 streams of the pedestrian. In the above case, the evacuation time required for 3000 people is $3000/(7 \times 43 \times 14) = 0.7$ (min), over about 42 s, which is consistent with the experimental evacuation time.

This simulation aims to select the individual with a personnel number of 500, extract the data of personnel density and speed, and draw the line chart as shown in Figure 24. In the program, the maximum speed of personnel is set as 1.5 m/s. It can be seen from the figure that the density of personnel is negatively related to the maximum speed of walking. In the period of 1–8 s, because the density of personnel is relatively small, the person can move to the exit direction at their maximum speed. In the period of 9–14 s, due to a large number of evacuees gathering at the exit, the density of personnel becomes large, and there is no moving space, along with the decreasing walking speed.

The relationship between the total number of evacuees at each exit and the time is shown in Figure 25. The maximum time of the exit flow is t = 5 s, and then the time outlet flow decreases continuously. An exit on the longer exterior wall of the evacuation scene can be selected, and the exit flow can be counted at different times, as shown in Figure 26.

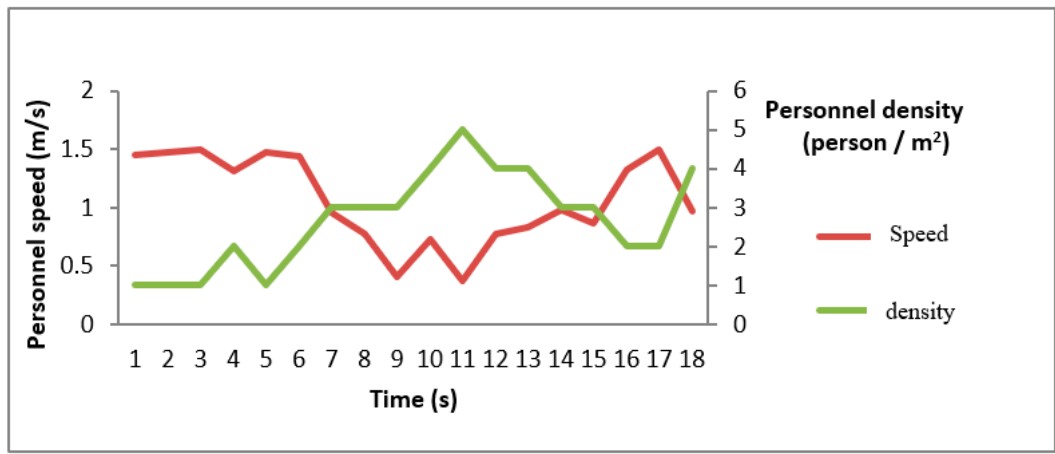

**Figure 24.** Personnel speed and density diagram.

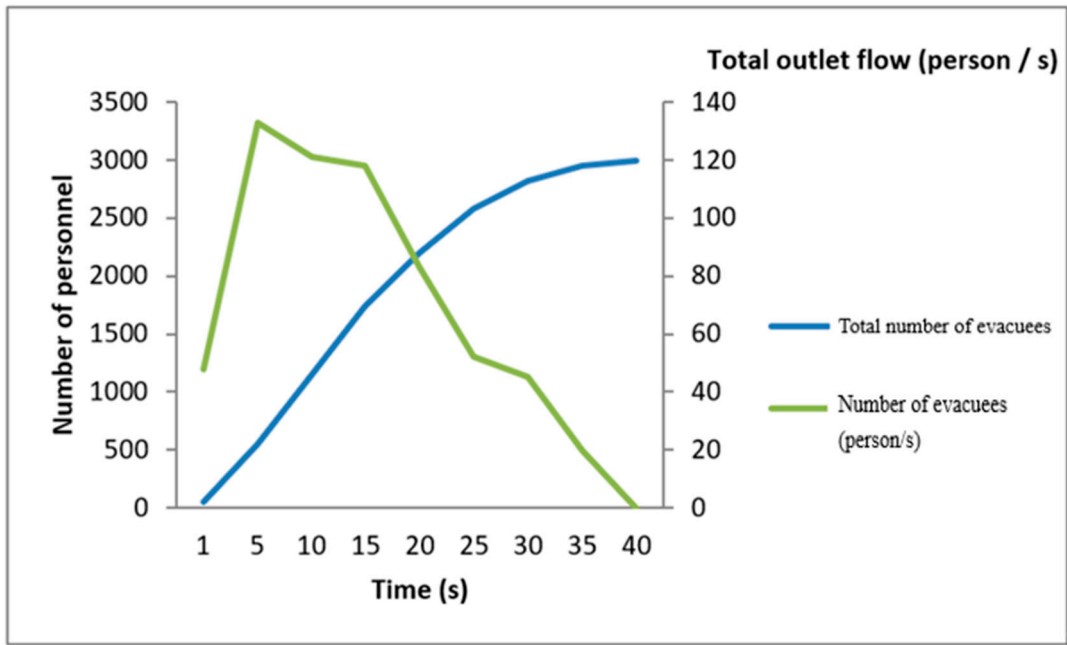

**Figure 25.** Relationship between evacuation number and time.

According to the state of pedestrian gathering at the entrance, the model can simulate the phenomenon of crowd gathering when waiting at the exit. For example, the exit with a symmetrical initial population distribution forms a symmetrical arc-shaped aggregation circle, while at the exit with asymmetric initial population distribution, the crowd tends to gather in the direction with more incoming people.

### 4.3.2. Simulation Comparison of Massmotion Software

Figure 27 shows the simulation results using Massmotion software developed by the OASYS company, which is based on the principle of the social force model.

The evacuation time simulated by Massmotion is more than 400 s. The results of the two are quite different. MassMotion takes longer because a large number of people are congested in the above figure. However, the intelligent evacuation model adopted by the software is not enough. Only a small number of evacuees in the congestion choose other exits, and most of them are waiting, which should be the initial design aim of personnel path planning, though it has little to do with the most basic motion model.

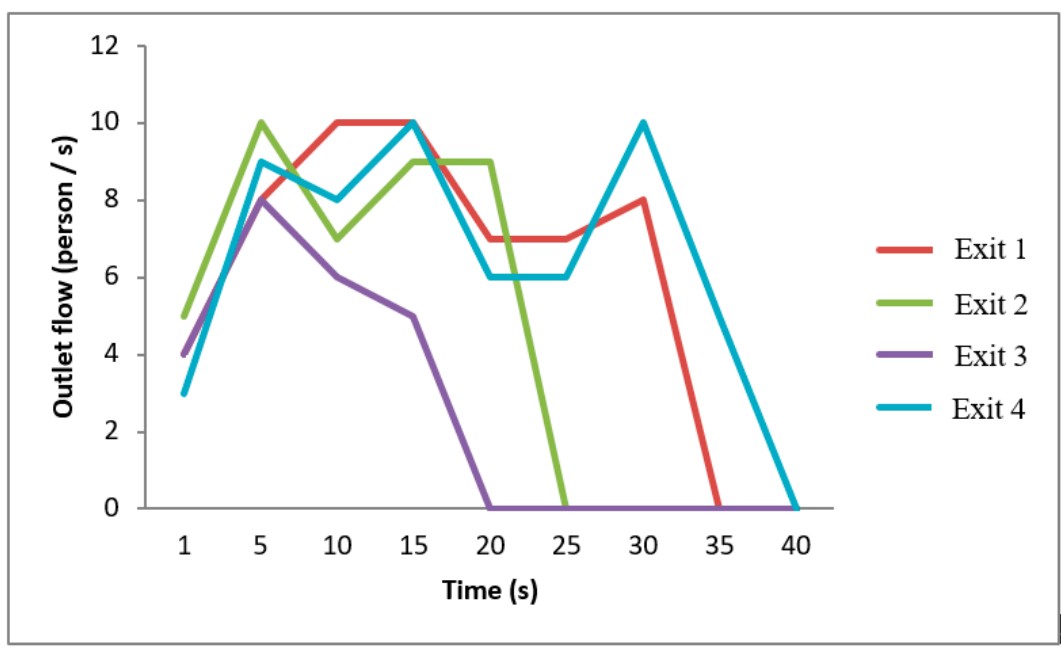

**Figure 26.** Outlet flow chart.

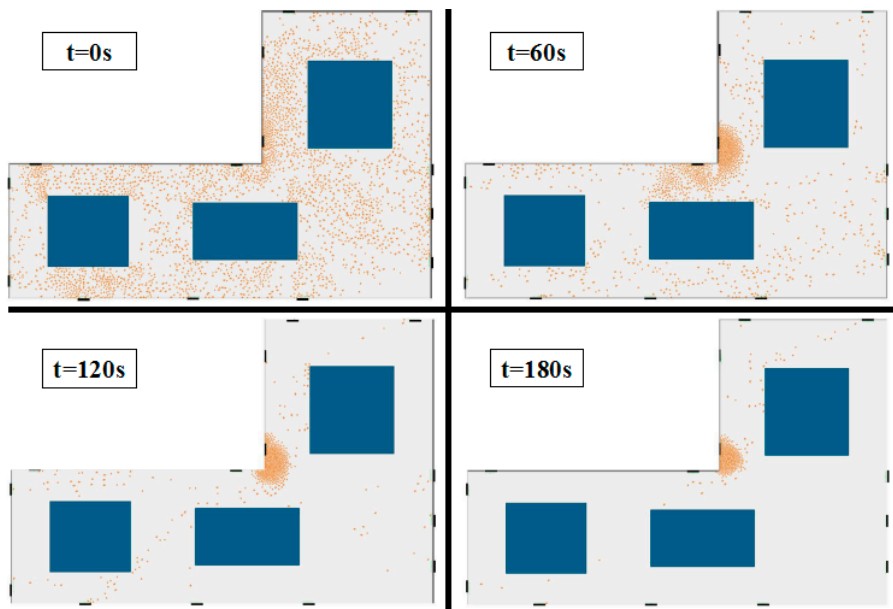

**Figure 27.** Screenshot of Massmotion software simulation.

4.3.3. Simulation Comparison of Experiments

In order to verify the validity of the model, we chose to measure the evacuation capacity of an evacuation site determined by the exit flow rate. In order to verify the validity of the model, we choose to measure the exit flow rate of the real place and compare it with the simulation of the evacuation model.

We propose selecting a classroom in a university teaching building, starting from the end of the last class in the morning. The students' walking speed is faster and reaches the average value of 1.5 m/s. The exit of measured data is located at the back of the classroom, with an exit width of 85 cm and a total number of 20 people. Both experiments are completed within 6 s. The experiment is conducted twice in total at different times, the evacuation video screenshots of which are shown in Figure 28. The outlet flow rates at different moments are shown in Figure 29.

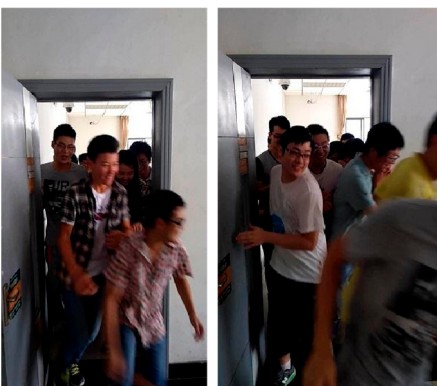

**Figure 28.** Screenshot of outlet flow experiment.

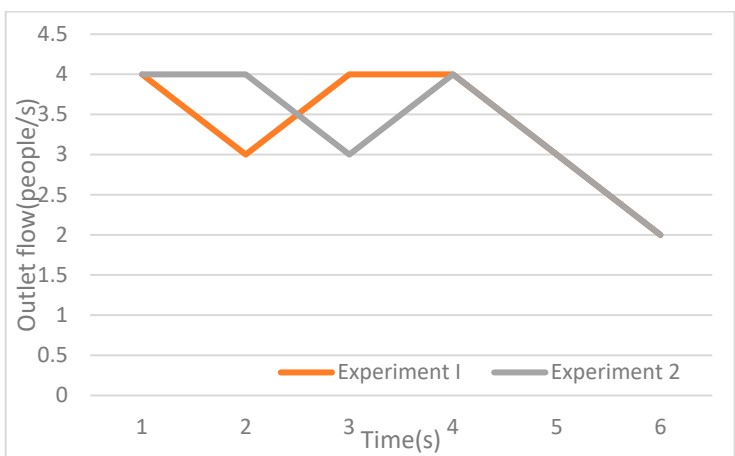

**Figure 29.** Line chart of outlet flow.

From the above figure, we can see that the exit flow can reach an average of 3 people/s. The scenario shown in the video is set in the program, whereby each parameter corresponds to the scene above, and the people are numbered. The screenshot of the evacuation simulation is shown in Figure 30.

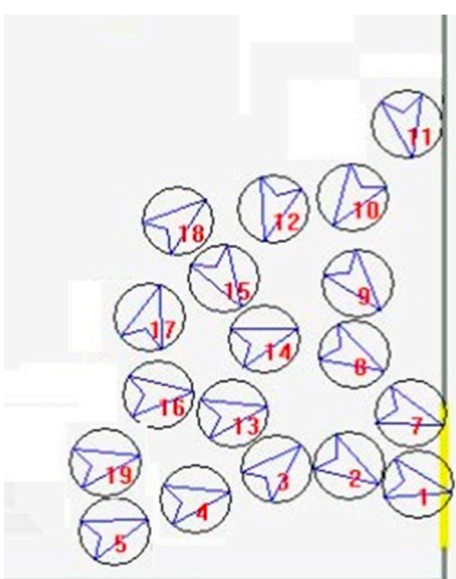

**Figure 30.** Line chart of outlet flow.

After several simulations using the program, the total time difference is not large, and three of the data sets are selected (as shown in Figure 31), and the mean value of the exit flow is in the range of 2–3 people/s, which is 1 person/s smaller compared to the experimental result of the exit flow. The analysis shows that the evacuation model treats individuals as circles, which is slightly wrong compared to the approximate elliptical fit between people. The experiment also shows that the people in the experiment rush out of the exit one after another at the maximum rate, and the people in the program take the exit as the arrival. The instantaneous speed of passing through the door is smaller than the former, based on the above two points caused the gap between the experiment and the simulation. The model plausibility is thus validated.

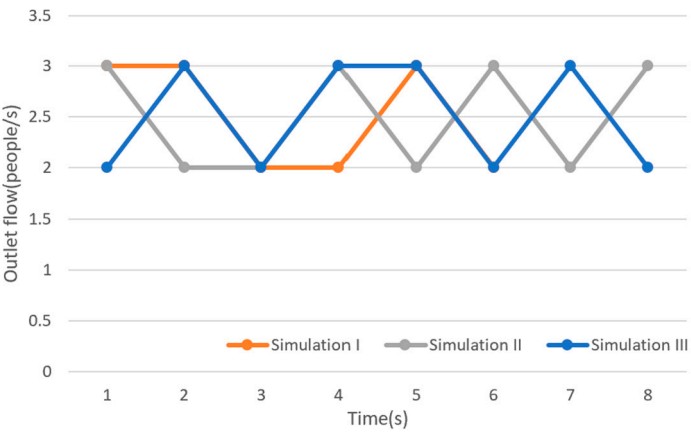

**Figure 31.** Line chart of the outlet flow simulation.

## 5. Conclusions

This paper constructs an agent-based evacuation model with the basic idea of "detection-decision-action". The dynamic equation of the model is based on Newton's second law of motion, with the implication of the decision-making model and the probability model to simulate the typical behaviors of avoiding pedestrians and obstacles in an evacuation, which mainly focuses on the walking force, as a combination action of the driving force, the repulsion force, and the steering force. Different from the social force model, the steering force mainly depends on the detection results while walking, and the steering angle is determined by the obstacle avoidance; once decided, it will be applied actively. The decision-making model is introduced to determine the walking strategy when encountering obstacles and other pedestrians. Thanks to the introduction of the algorithm of steering force application and the decision-making model, the AEM proposed in this paper is more intelligent than the previously implemented models.

The research theorizes the intelligence algorithms of detection and obstacle avoidance which express evacuees' behaviors such as following or surpassing other evacuees and avoiding obstacles in the evacuation process, which can also be applied into the scene of traffic evacuation to demonstrate the distribution of evacuees and the surpassing behavior while walking, thus smoothening the whole evacuation process even in complicated structured areas with narrow passageways. Furthermore, this research simulates an evacuation in a complex place, taking an exhibition center as the example.

The experimental results show that, compared with the previous grid model and the social force model, the AEM solves the problem of pedestrian evacuation behavior distortion to a certain extent, which also can deal with dense obstacle scenes, making pedestrians more flexible in evacuation path selection.

**Author Contributions:** Conceptualization, X.C. and J.J.; methodology, X.C.; software, X.C.; validation, X.C., J.J. and X.B.; investigation, X.B.; data curation, X.C.; writing—original draft preparation, X.C.; writing—review and editing, J.J.; visualization, X.C.; supervision, J.J.; project administration, J.J. All authors have read and agreed to the published version of the manuscript.

**Funding:** This research received no external funding.

**Informed Consent Statement:** Not applicable.

**Data Availability Statement:** No new data were created or analyzed in this study. Data sharing is not applicable to this article.

**Conflicts of Interest:** The authors declare no conflict of interest.

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
