# Peer review of "Algorithm and Examples of an Agent-Based Evacuation Model"

_fire, doi:10.3390/fire6010011_

Round 1
Reviewer 1 Report
Please answer according to the question number.
Q1: The article must be amended to include up to two reference numbers when creating reference numbers. This measure conveys specific information to the reader precisely. References 2 to 8 correspond to this.
Q2: Page 2 of the text, 4th column from the bottom: It is necessary to explain what 'real virtual entity object' means in detail.
Q3: Page 4 of the text, 12th column from the bottom: Equation 1.2 à Check the equation number.
Q4: Page 4 of the text, 4th column from the bottom: ‘The driving force is determined as Equation 1.4:n’ à Check the equation number.
Q5: Equation (3) requires the entry of supplementary formulas omitted for better understanding. This effort increases the reader's readability as much as possible.
Q6: A description of the parameters written to the equations is essential. This applies to all equations in the article. To better understand the kinematics model, the equations described in this article should be written so they can be clearly understood. This is because it occupies a significant part in evaluating the theory asserted in this article.
Q7: Page 5 of the text, 1st column from the top: ‘The repulsive force can be described as Equation 1.5:’à Check the equation number.
Q8: Please add a picture with vectors to explain the equations for the kinematic model in 3.1 for better readability
Q9: Explaining the equations for the kinematic model in 3.1 is recommended to improve readability by adding a picture containing vectors to enhance understanding.
Q10: In Figure 5, fixed length detection radius r2 related to visibility needs more detail to explain its function.
Q11: Page 8 of the text, 14th~ 26th column from the bottom: It is difficult for readers to understand the connection of expressions based on the contents of the text, so special efforts are required to increase understanding. Are the indicated variables A and C the descriptions displayed associated with the items of equations 9 and 10? Therefore, it is necessary to supplement the described content to a level the reader can fully understand.
Q12: In Fig.6, r1 should be the notation from the center of object A.
Q13: Text page 11, column 20 from the bottom; This article introduced probability ? to assist make evacuation decisions. A detailed explanation of how probability ? was applied is needed.
Q14: In the obstacle avoidance algorithm, the repulsive force between the pedestrian and the wall is intended for general avoidance. However, the frictional force between the pedestrian and the wall must act in the close-packed movement. Therefore, it is thought that this resistance force affects the speed of the pedestrian. What is the author's opinion on the need for this?
Q15: When the author interpreted it as a condition for one-way dense movement conditions in this study, does the application of repulsive force between pedestrians act as a factor in increasing walking speed?
Q16: In the application research of a new theory, comparing the difference with the existing approach by interpreting it as a simulation under the same conditions is an excellent way to confirm the author's originality rather than introducing various conditions' results according to the author's method. Therefore, a comparative analysis with the current research for the same evacuation conditions is essential. This data supports the conclusion that excellent interpretation is possible due to the application of the steering force in the final decision, and it must be included in the scope of this article.
/END
Author Response
Dear Editor:
The attachment contains the reply and the revised original manuscript.
Best wishes!

Reviewer 2 Report
Summary
The proposed model for evacuation simulations is a hybrid approach that combines elements of mechanical modeling with decision-making based on social and personal behaviors. The use of an agent-based approach allows for the simulation of individual agents and their interactions with each other and the environment, which can be useful for understanding the complex dynamics of evacuation situations. The inclusion of mechanical considerations such as obstacle detection and avoidance can help to make the model more realistic, while the decision-making algorithm based on social and personal behaviors can help to capture the diverse behaviors of individuals in emergency situations. The discussion of the results and their implications for the development of more effective evacuation models is also a valuable aspect of the research, as it helps to contextualize the findings and suggest potential areas for future research.
Major Issues
There are some major issues with the paper - specifically, the paper lacks a general overview of the methodology followed, and the conclusions are not supported by the results of the simulation. Additionally, there is no comparison with the results from other methodologies, such as the Social Force model or Grid models, which makes it difficult to determine if the new methodology is more effective than existing ones.
To address these issues, it would be helpful for the authors to provide a more comprehensive overview of the methodology followed in the paper, including a literature review and a clear illustration of the method. This would allow readers to understand the basis for the new methodology and how it differs from existing approaches. Additionally, the authors should include a more thorough comparison of the results from their new methodology with those of other models, such as the Social Force model and Grid models, in order to demonstrate the relative effectiveness of their approach. Moreover, the authors should provide more information on why they selected the Massmotion software for comparison and whether it is based on social force or grid models.
Minor Issues
These are some minor issues that need to be addressed in the paper to improve its clarity and coherence.
First, it would be helpful for the authors to better illustrate the literature gap and explicitly identify the advantages of the newly proposed method in order to clearly communicate the significance of their work.
Second, there are a few issues with the writing that could be revised for clarity. In line 1, the phrase "the research is guided the process" is unclear and should be rewritten. In line 6, the acronym AEM should be explained. In line 14, the last sentence does not seem to be connected to the previous one and should be rewritten. In line 52, the phrase "discrete models by dividing" is unclear and should be revised.
Third, there are a few places in the paper where more information or clarification is needed. In line 314, it is not clear how the aggressive attribute is included in the model. In line 331, references for safety ergonomics should be provided. In line 601, the results provided graphically in figure 12 are confusing, and it is not clear what the outcome of the methodology is or what is being measured in the velocity and detection range. In line 720, the reference is not clear. In line 721, the concept of "passing capacity of per capita flow" is not clear and should be rewritten. In line 725, the subject is missing in these sentences. In line 747, a capital letter should be used and a reference to the Massmotion software should be added. Finally, section 4.3.1 is poorly written and should be expanded.
Finally, it would be helpful for the authors to include a comparison with the Social Force model or the Grid model in their paper, as these models are not mentioned in the text. It is also not clear if these models are included in the Massmotion algorithm.
Please ensure that the English verbiage is checked properly, there are too many grammatical errors which is just not acceptable for a high quality Fire journal. We do not want to the laughing stock of the Fire industry on account of our poor grammatical English.
Author Response
Dear Reviewers.
The attached document contains the responses to your suggestions and the revised original manuscript for your review.
Best wishes!

Reviewer 3 Report
This paper developed an Agent-based Evacuation Model based on the "detection-decision-action" approach. The dynamic equation of the model is based on Newton's second law of motion, with the implication of the Decision-Making Model and Probability Model to simulate the typical behaviors of avoiding pedestrians and obstacles in an evacuation situation. The Decision-making Model is introduced to determine the walking strategy when encountering obstacles and other pedestrians. The authors compare their approach with others developed by the OASYS company The paper is generally well structured and has fresh ideas .it can improve in writing, and there are things that can be explained and detailed better. My suggestions are the following:
-Fig 11. It is not explained or modeled how trajectory is generated. Please elaborate on the specific movement details.
-How are the simulation of figures 15, 16, and 17 done? in what software?
-How is the simulation of Figure 22 done? in what software?
These recommendations must be resolved for this paper to be published.
Author Response
Dear Reviewers.
Attached is the response to your suggestion, please check it.
Best wishes!

Reviewer 4 Report
This paper proposes an agent-based evacuation model based on interactions between individuals as well as between an individual and obstacles. Illustrative examples have been provided on the simulation results for different scenarios, including an actual scene for an exhibition center. However, it is not clear how the model proposed in this study is different from other agent-based models that are available for academic purpose or for commercial use in the literature. The contribution of the presented method to fill existing research gap should be stated clearly at the onset of the manuscript. Moreover, as several assumptions have been employed, the efficacy of the model needs to be verified based on empirical data or by comparing the results with those obtained from other models in the literature under the same scenarios. Without this information, the paper would serve better as an academic example rather than a research article. Therefore, it is the reviewer’s opinion that the manuscript in its present form is not suitable for publication as a research article in Fire.
Author Response
Dear Reviewers.
Attached is a response to your suggestion, as well as the revised original manuscript, for your review.
Best wishes!

Round 2
Reviewer 1 Report
You should submit your final manuscript after improving your understanding of complex sentences and final confirmation of English sentences.